# Reprocessable Polybenzoxazine Thermosets with High T_g_s and Mechanical Strength Retentions Using Boronic Ester Bonds as Crosslinkages

**DOI:** 10.3390/polym14112234

**Published:** 2022-05-31

**Authors:** Xiaoxi Wang, Sujuan Zhang, Youjun He, Wei Guo, Zaijun Lu

**Affiliations:** Key Laboratory for Special Functional Aggregated Materials of Ministry of Education, School of Chemistry and Chemical Engineering, Shandong University, Jinan 250100, China; 18763131501@163.com (X.W.); zsjlab601@163.com (S.Z.); hyjlab601@163.com (Y.H.); gwlab601@163.com (W.G.)

**Keywords:** polybenzoxazine, boronic ester bond, reprocessability, heat-resistance

## Abstract

In order to obtain reprocessable polybenzoxazine thermosets with high heat resistance and mechanical strength retentions, network structures without irreversible parts were constructed via crosslinking benzoxazine oligomers using boronic ester cross-linkers. Firstly, the benzoxazine monomer containing carbon–carbon double bonds was synthesized via the Mannich reaction. After thermal ring-opening polymerization, the benzoxazine oligomer containing carbon–carbon double bonds (OBZ) was yielded. Through the thiol-ene click reaction of the OBZ and dithiol cross-linker bearing boronic ester bonds, the polybenzoxazine thermosets using boronic ester bonds as crosslinkages (OBZ-BDB) were successfully synthesized. The structures of OBZ and OBZ-BDB were characterized by SEC, ^1^H NMR, and FT-IR measurements. Reprocessing experiments showed that OBZ-BDB has remarkable reprocessability. The retention rates of the tensile strengths through three generations of reprocessing were 98%, 95%, and 84%, respectively. Meanwhile, OBZ-BDB cross-linked by boronic ester bonds had brilliant thermal properties. The T_g_ of the original OBZ-BDB was 224 °C. With the increase of the reprocessing generations, the T_g_s basically remained unchanged.

## 1. Introduction

Thermosetting polymers have been widely used in our daily life because of their high mechanical strengths, good chemical resistance, and excellent dimensional stability [1]. However, owing to their irreversibly cross-linked structures, traditional thermosetting polymers cannot be dissolved, melted, and reprocessed, resulting in the wasting of resources and environmental pollution. Recently, the incorporation of dynamic covalent bonds into polymer networks was a breakthrough to solve this problem. Through the reversible breaking and formation of dynamic covalent bonds, those dynamic cross-linked polymers can be reprocessed [2,3,4]. Nevertheless, the synthesis of reprocessable thermosetting polymers with high T_g_s and retentions of mechanical strengths has still been a challenge so far.

Generally, dynamic covalent bonds can be classified into the dissociative type and the associative type according to their exchange mechanisms [5]. In the former, the bond breaks first and then reforms after some time, as in a Diels-Alder addition reaction [6]. In the latter, bond breaking and reformation take place simultaneously, such as in disulfide bonds [7], ester bonds [8], boronic ester bonds [9,10], urea bonds [11], and acylhydrazone bonds [12]. Therefore, for thermosetting polymers containing an associative dynamic covalent bond, there is no change to the crosslinking density during the exchange process, and the overall structure of the thermosetting polymers is rarely damaged even at high temperatures.

Benzoxazine resin, as a new class of thermosetting polymer, has gained considerable attention in recent years on account of its brilliant heat resistance, excellent flame retardance, and high hydrophobicity [13,14,15]. Recently, various dynamic covalent bonds have been introduced into polybenzoxazine networks to achieve self-healabilty or reprocessability [16,17,18,19,20,21,22,23,24]. However, there have been few reports on introducing boronic ester bonds into benzoxazine resins until now. That research mainly focused on thermal properties [25,26,27,28], toughness modification [29], and fluorescence modification [30]. Reprocessable research via boronic ester bonds is just beginning [31].

Considering that boronic ester bonds can dramatically improve the thermal stability due to their high bond energy [32], and that they can easily construct a rigid polymer matrix to achieve high T_g_, dynamic boronic ester bonds were chosen to synthesize reprocessable benzoxazine resin (OBZ-BDB) with high T_g_. In order to achieve the high retention of mechanical strength, benzoxazine oligomers were designed as the skeleton chain. It was further reacted with boronic ester cross-linkers to construct a purely reversible cross-linking network instead of a network with irreversible parts. Specifically, the benzoxazine monomer containing carbon–carbon double bonds was synthesized via the Mannich reaction. After thermal ring-opening polymerization, a benzoxazine oligomer bearing a carbon–carbon double bond was synthesized. Finally, we utilized the thiol-ene click reaction between the carbon–carbon double bonds of the benzoxazine oligomer and the thiols of boronic ester cross-linker to produce the reprocessable polybenzoxazine of OBZ-BDB.

## 2. Experimental Section

### 2.1. Materials

Allylamine (99.5%) was purchased from Zouping Mingyuan Import and Export Trade Co., Ltd. (Binzhou, China). Cardanol (98%) was obtained from Jinquan Chemical Co., Ltd. (Jinan, China). Paraformaldehyde (analytically pure) and neutral aluminium oxide (analytically pure) were purchased from Sinopharm Chemical Reagent Co., Ltd. (Shanghai, China). Furthermore, 1-Thioglycerol (95%) was purchased from J&K Chemical Ltd. (Beijing, China). Benzene-1,4-diboronic acid (98%) was supplied by Shanghai Bide Medical Technology Co., Ltd. (Shanghai, China). P-*tert*-butylphenol (analytically pure) and anisole (99%) were obtained from Shanghai Macklin Biochemical Co., Ltd. (Shanghai, China). Azobisisobutyronitrile (AIBN, 99%) was obtained from J&K Chemical Ltd. (Beijing, China). and purified by recrystallization from methanol. Toluene, *n*-hexane, methanol, and ethanol were purchased from Fuyu Fine Chemical Co., Ltd. (Tianjin, China). All of the other reagents and solvents were used as received.

### 2.2. Synthesis of Benzoxazine (BZ)

To a three-neck round flask, allylamine (8.55 g, 150 mmol), paraformaldehyde (9.47 g, 300 mmol), cardanol (46.61 g, 150 mmol), and toluene (300 mL) were added and refluxed for 12 h. Then, neutral alumina was added and stirred for 0.5 h. The product was collected by filtration and evaporated to remove toluene. The solution was collected by filtration and dropped into cold methanol. Then, the lower phase was collected. After being dried in vacuum at 60 °C for 24 h, a yellowish liquid was obtained (yield 83%).

^1^H NMR (400 MHz, CDCl_3_, ppm): 6.90–6.60 (3H, Ar–***H***), 5.96–5.84 (1H, –CH_2_–C***H***=CH_2_), 4.85 (2H, O–C***H***_2_–N), 5.28–5.15 (2H, C***H***_2_=CH–), 3.96 (2H, Ar–C***H***_2_–N), 3.41–3.35 (2H, N–C***H***_2_–CH–), 2.57–2.48 (2H, Ar–C***H***_2_–), and 1.64–1.51 (2H, Ar–CH_2_–C***H***_2_–CH_2_–).

FT-IR (KBr, cm^−1^): 2926 and 2854 (CH_2_ of the oxazine ring as well as the alkyl side chain of cardanol), 1505 (tri-substituted benzene ring mode), 1241 (C–O–C of the oxazine ring), and 968 (the oxazine ring mode).

### 2.3. Synthesis of Benzoxazine Oligomer (OBZ)

P-*tert*-butylphenol (1.5 g) was added to BZ (15 g) and stirred at 100 °C for 5 min. The mixture was heated at 150 °C for 10 h. After cooling to room temperature, the crude product was dissolved in *n*-hexane and added dropwise into methanol for precipitation. After filtration and being dried in a vacuum at 50 °C for 12 h, an orange product was obtained (yield 70%).

^1^H NMR (400 MHz, CDCl_3_, ppm): 7.20–6.45 (Ar–***H***), 6.10–5.88 (–CH_2_–C***H***=CH_2_), 5.28–5.11 (C***H***_2_=CH–), 4.06–3.54 (Ar–C***H***_2_–N), 3.41–3.35 (N–C***H***_2_–CH–), 2.51 (Ar–C***H***_2_–), 1.68–1.45 (Ar–CH_2_–C***H***_2_–CH_2_–), and 0.89 (C***H***_2_–CH_2_–CH_2_–CH_2_).

FT-IR (KBr, cm^−1^): 2926 and 2854 (CH_2_ of the oxazine ring as well as the alkyl side chain of cardanol), and 1483 (tetra-substituted benzene ring mode).

### 2.4. Synthesis of 2,2′-(1,4-Phenylene)-bis [4-mercaptan-1,3,2-dioxaborolane] (BDB)

BDB was synthesized according to the literature [33,34]. Benzene-1,4-diboronic acid (3.03 g, 18.1 mmol) and 1-thioglycerol (4.12 g, 36.2 mmol) were dissolved in ethanol (65 mL) and stirred at room temperature for 24 h. After being evaporated to remove ethanol, a white solid was obtained (yield 92%).

^1^H NMR (400 MHz, CDCl_3_, ppm, Appendix A): 1.48 (2H, –S***H***), 2.81 (4H, HS–C***H***_2_–), 4.18 and 4.49 (4H, O–C***H***_2_–), 4.74 (2H, O–C***H***–(CH_2_–)_2_), and 7.83 (4H, Ar–***H***).

FT-IR (KBr, cm^−1^): 2568 (–SH stretching vibration), 1219 (B–O stretching vibration), and 656 (B–O–B linkages within borate networks).

### 2.5. Preparation of Polybenzoxazine Cross-Linked by the Boronic Ester Bond (OBZ-BDB)

OBZ (3.02 g, 9.94 mmol of the vinyl groups, as calculated by ^1^H NMR), BDB (1.51 g, 4.87 mmol), and AIBN (0.13 g, 0.79 mmol) were dissolved in 5 mL anisole to form the solution. Subsequently, the solution was poured into a Teflon mold. The sample was thermally cured according to the following temperature sequence: 75 °C for 6 h, 100 °C for 2 h, 120 °C for 2 h, and 140 °C for 2 h. Finally, it was dried under vacuum at 100 °C for 24 h.

### 2.6. Characterization

Nuclear magnetic resonance (NMR): ^1^H NMR spectra were recorded on an AVANCE III HD 400 MHz NMR spectrometer (Bruker, Romanshorn, Switzerland) with CDCl_3_ as the deuterium solvent.

Fourier transform infrared spectroscopy (FT-IR): FT-IR spectra were obtained on a TENSOR 27 FTIR spectrometer (Bruker, Romanshorn, Switzerland) with a resolution of 4 cm^−1^ across the 4000–500 cm^−1^ range by the KBr pellet method.

Size exclusion chromatography (SEC): The number average molecular weight and polydispersity index were measured by SEC using a 515 liquid chromatograph equipped with 3 Styragel columns and a refractive-index detector (Waters, Milford, MA, USA) in tetrahydrofuran (1 mL·min^−1^) at 40 °C. Polystyrene standards were used for calibration.

Dynamic mechanical analysis (DMA): DMA experiments were performed on a DMA/SDTA 861^e^  instrument (Mettler-Toledo, Zurich, Switzerland) in shear mode. A specimen with dimensions of approximately 5.0 × 5.0 × 0.5 mm^3^ was tested at 1 Hz and 3 μm oscillation amplitude. The temperature was increased from 0 °C to 290 °C at a heating rate of 3 K·min^−1^.

Thermo-gravimetric analysis (TGA): TGA was performed on a TGA/DSC STAR^e^ instrument (Mettler-Toledo, Zurich, Switzerland) under a nitrogen flow at a heating rate of 10 °C·min^−1^, from 30 °C to 800 °C.

Tensile tests: The tensile tests were performed on a testing machine UTM5105 (Instron, Shanghai, China) at a crosshead speed of 2 mm·min^−1^. Each sample was tested respectively for five specimens.

Reprocessing tests: The OBZ-BDB samples were ground into powder and fully filled into stainless steel mold. The dumbbell-shaped reprocessed samples were obtained by hot pressing under 16 MPa at 160 °C for 2 h. Each sample was tested respectively for five specimens.

## 3. Results and Discussion

### 3.1. Synthesis of OBZ-BDB

Figure 1 illustrates the synthetic route to the polybenzoxazine OBZ-BDB cross-linked by boronic ester bonds. Allylamine, paraformaldehyde, and cardanol were chosen as raw materials to yield a BZ monomer containing C=C via the Mannich reaction in one pot. Here, allylamine was selected as an amine source because it has a C=C bond and is suitable for the thiol-ene click reaction. Meanwhile, there are two main reasons to select cardanol as a phenol source. One is that the cardanol contains a terminal C=C bond, which can provide more cross-linking sites for the thiol-ene click reaction. The other is that the cardanol can be easily obtained from cashew nut shells, an agricultural waste product. After the BZ monomer underwent thermal ring-opening polymerization at 150 °C, the OBZ oligomer bearing a C=C bond was yielded. According to the literature [33,34], boronic ester cross-linker (BDB) containing thiols were synthesized through a condensation reaction. Finally, OBZ reacted with BDB via the thiol-ene click reaction to synthesize the cross-linked OBZ-BDB.

Figure 1 shows the SEC chromatograms of OBZ and BZ. Compared with BZ, the elution time of OBZ moves to a higher molecular weight region, indicating the synthesis of the OBZ oligomer. The number-averaged molecular weight of OBZ is 1900 g mol^−1^ and its polydispersity index is 3.50.

Figure 2 shows the ^1^H NMR spectrum of OBZ (d). For comparison, allylamine (a), cardanol (b), and BZ monomer (c) are also listed. For BZ, the characteristic resonances at 4.85, 3.96, and 3.39 ppm are attributed to the protons of O–C***H***_2_–N (9), Ar–C***H***_2_–N (10), and N–C***H***_2_–CH=CH_2_ (2), respectively. The ratio of the integral areas of these resonance signals is 2:2:2, which is in good agreement with the theoretical value. This indicates the successful synthesis of the BZ monomer. For OBZ, the characteristic peaks of the oxazine ring at 4.85 and 3.96 ppm completely disappear. Meanwhile, a new characteristic peak corresponding to the Mannich bridge protons of –C***H***_2_–N(R)–C***H***_2_– (2) appears at 3.72 ppm, indicating the ring-opening polymerization of BZ. This is also confirmed in the FT-IR spectra of OBZ (see Appendix A). Compared with BZ, the absorption peak at 968 cm^−1^ assigned to the oxazine ring disappears completely, and a new absorption peak at 1483 cm^−1^ assigned to tetra-substituted benzene appears, which further suggests the successful ring-opening polymerization of BZ.

In order to construct an ideal network, the molar ratio of alkene to thiol is a crucial factor. If the crosslinking agent (BDB) is insufficient or excessive, some crosslinking points would not be formed. Therefore, we used an equal molar ratio to feed the alkene and thiol. The calculated molar content of CH_2_=CH– is 1.24 mol per one mol of the repeat unit of OBZ according to the following equation:(1)$$n_{{–\mathrm{CH}=\mathrm{CH}}_{2}}=2\left(A_{5.97}{+A}_{5.81}\right)/A_{2.51}$$
where A_5.97_, A_5.81_, and A_2.51_ are the integral areas of the ^1^H NMR signals for CH_2_=C***H***-CH_2_-N (3), CH_2_=C***H***-CH_2_-CH (b_2_), and Ar-C***H***_2_-CH_2_ (o), respectively. The calculated feed mass ratio of $m_{\mathrm{BDB}}/m_{\mathrm{OBZ}}$ was 50.2%.

Figure 3 shows the FTIR spectra of OBZ-BDB before (a) and after (b) curing. By comparison, the absorption peak at 2568 cm^−1^ ascribed to –SH disappears after curing (see Figure 3b). Moreover, the peak intensity at 914 cm^−1^ assigned to the terminal vinyl bond obviously weakens. However, the peak intensity at 729 cm^−1^ assigned to the inner vinyl group in the cardanol side chain has little change. These results indicate that the click reaction mainly occurs in allyl double bonds and the terminal double bonds of the cardanol side chain.

### 3.2. Reprocessability of OBZ-BDB

Figure 4 shows digital photographs of the reprocessing process and the reprocessing mechanism of OBZ-BDB. The original transparent OBZ-BDB film was ground into powder and then filled into the dumbbell-shaped molds. After heating and pressurizing, a transparent and uniform dumbbell-shaped sample was obtained.

We think that the reprocessing mechanism of OBZ-BDB is shown as follows. During the hot-pressing process, the boronic esters at the interfaces between ground particles produce a metathesis reaction, generating some new cross-linking sites across interfaces. This causes ground particles to fuse with each other to form a whole. Therefore, the appearance of the samples is transparent and uniform.

Figure 5a displays the influence of the hot-pressing temperature on the tensile strength. Obviously, with the increase of the hot-pressing temperature from 140 °C to 160 °C, the tensile strength gradually increases. When the temperature increases more than 160 °C, the tensile strength of reprocessed OBZ-BDB doesn’t change much. It is well-known that the realization of reprocessing requires dynamic covalent bonds. On the other hand, it also needs to be carried out in the viscoelastic state or the flow state. Therefore, when the hot-pressing temperature increases higher and higher, the movement of the chain segments is favorable, and the ground particles are prone to deformation and healing. This leads to higher tensile strengths.

Figure 5b shows the influence of the hot-pressing time on the tensile strength. It is clearly seen that the tensile strength increases from 17.8 MPa to 31 MPa with the extension of the hot-pressing time from 0.5 h to 2 h. When the hot-pressing time further increases to 3 h, the tensile strength of reprocessed OBZ-BDB hardly increases.

Figure 6 presents the tensile performance of the original OBZ-BDB, and after three generations of reprocessing. The related data are listed in Appendix A. The original tensile strength and elongation at the break of OBZ-BDB are 31 MPa and 8.4%, respectively. With the increasing of the reprocessing generation, the tensile strengths of reprocessed OBZ-BDB gradually decrease to 30.6, 29.6, and 26.0 MPa. Compared with the original OBZ-BDB, the retention rates of the tensile strengths after three reprocessing generations are 98%, 95%, and 84% respectively. In addition, OBZ-BDB still retains 76% of the original elongation at the break after the third reprocessing generation. These results suggest that OBZ-BDB has excellent reprocessability. We think this is mainly caused by the ideal cross-linked network structure we designed. Due to using benzoxazine oligomer as a chain backbone and boronic ester as a cross-linking agent, the dynamic covalent boronic ester bonds all exist at the cross-linking points. The formed cross-linked system does not contain irreversible networks or reversible/irreversible interpenetrating networks. This well-defined reversible network structure is beneficial to obtain a high retention rate of mechanical strength.

### 3.3. Thermal Properties of OBZ-BDB

Figure 7 exhibits the DMA curves of original and reprocessed OBZ-BDB. The corresponding T_g_s are listed in Table 1. The T_g_ of the original OBZ-BDB is 224 °C, indicating that OBZ-BDB has high heat-resistance. We think this is related to the high crosslinking density of OBZ-BDB. The segmental motions are frozen. The boronic ester bonds are embedded in a rigid polymer matrix, and their dynamic exchange reaction is inhibited. The OBZ-BDB exhibits high T_g_. With the increase of reprocessing generations, no significant change in T_g_s is observed, indicating that the reprocessed OBZ-BDB sample still has excellent heat resistance.

Figure 8 shows the TGA thermograms of original and reprocessed OBZ-BDB. It is easily noticeable that there are two main stages of weight loss. The first stage, ranging from 270 °C to 390 °C, is due to the decomposition of boronic ester bonds in BDB [34] and the cleavage of the phenolic Mannich bridges [35]. The second stage, ranging from 390 °C to 530 °C, is assigned to the degradation of the aliphatic side chain of the cardanol moiety, the phenolic moiety [35], and the remaining BDB moiety [34]. The related thermal performance data are presented in Table 1. The T_d5_, T_d10_, and char yield of the original OBZ-BDB remain at 325 °C, 341 °C, and 30%, suggesting that OBZ-BDB has favorable thermal stability. After three reprocessing generations, these parameters of OBZ-BDB can still remain at 312 °C, 328 °C, and 24%, indicating that the reprocessed OBZ-BDB also possesses good thermal stability.

## 4. Conclusions

In conclusion, a reprocessable benzoxazine resin of OBZ-BDB cross-linked by boronic ester bonds was successfully synthesized via a thiol-ene click reaction between boronic ester crosslinker containing thiols and an OBZ oligomer containing double bonds.

The hot-pressing temperature and time have an obvious influence on the tensile strength. The optimized reprocessing conditions are hot-pressing at 160 °C under 16 MPa for 2 h.

The reprocessing tests confirm that the OBZ-BDB has remarkable reprocessability. The retention rates of the tensile strength through three generations of reprocessing are 98%, 95%, and 84%, respectively.

The OBZ-BDB has excellent heat resistance. The T_g_ of the original OBZ-BDB is 224 °C. After three generations of reprocessing, no significant change in T_g_s was observed.

## Data Availability

Data is contained within the article and Appendix A.

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
