# Peer review of "Reprocessable Polybenzoxazine Thermosets with High Tgs and Mechanical Strength Retentions Using Boronic Ester Bonds as Crosslinkages"

_polymers, 2022, doi:10.3390/polym14112234_

Round 1
Reviewer 1 Report
The manuscript describes the synthesis and characterization of reprocessable polybenzoxazine crosslinked via boronic ester bonds. Benzozaxine oligomers are successfully synthesized and characterized though NMR, FTIR and SEC. Thiol-containing boronic esters are used as crosslinker via thiol-ene reaction. The choice of boronic esters provide reversible covalent bond and ultimately reprocessability properties. Moreover, the thermosets shows beneficial thermal properties and heat resistance.
Overall, the flow of the manuscript is very good and easy to read, the experimental part is for the most part complete, and goal and results are clearly stated. However, the manuscript would benefit from minor editing and English check.
More detailed comments and suggestions can be found below:
- Section 2.3: Missing NMR string of OBZ. It can still be obtained also for oligomers.
- Section 2.5: Include mol or equivalents (stoichiometric ratio) of reagents. SEC molecular weight can be used for the calculations.
- It would be useful to report and discuss the alkene:thiol ratio used for the curing and thermoset preparation. If too many BDB molecules are used in the reaction, the chance of a BDB molecule to react with 2 different polymer chain and crosslink them become less likely, as monofunctionalization is prevalent. I think the authors should add a short discussion on this aspect.
Reviewer 2 Report
Dear authors, in general the mnuscript is nice to read and well presented.
There is an article, related to your studies, which surprisingly has not been cited: J. Mater. Chem. A, 2019,7, 17498-17504. Reprocessable and degradable thermoset with high Tg cross-linked via Si–O–Ph bonds. As a recommendation, I think it should be added in your introduction.
For the rest, some minor English corrections are needed. For example in line 207 the "way of" writing doesn't sound right. Shouldn't it be "it is clearly found"?
In line 230 it is related or this is related?
in lines 231 and 232: the boronic ester bonds are (so this is plural) and therefore in the next line their dinamic exchange...
line 240: it is easily noticeable that...
So in general I would advice to check the English grammar by a native speaker.
No further corrections are required from my side.
Reviewer 3 Report
In order to obtain the reprocess-able polybenzoxazine thermosets with high heat-resistance and mechanical strength retentions, the network structures without irreversible parts have been constructed via crosslinking benzoxazine oligomers using boronic ester cross-linkers. The cross-linked polymer has good thermal properties with Tg of 224℃. With increase of reprocessing generations, the Tg basically remained unchanged. The paper is interesting and could be published after revision. -Benzoxazine (BZ)was not purified after synthesis ? What was purity of the material ? Mass spectrum should be provided for the compound. - 2,2′-(1,4-Phenylene)-bis[4-mercaptan-1,3,2-dioxaborolane] (BDB) should be characterized by mass spectroscopy. What was purity of the material after synthesis ? - Preparation of polybenzoxazine cross-linked by boronic ester bond (OBZ-BDB): The sample was thermally cured according to the following temperature sequence: 75℃ for 6 h, 100℃ for 107 2 h, 120℃ for 2 h, 140℃ for 2 h. Why such complicated sequence was used ?? -Synthesis is rather complicated. Would the product have an industrial application economically ? -Titles in some figures should be normalized. -Advantages and disadvantages of the developed polymer should be compared in conclusions with that of other thermosetting polymers, which are described in literature.Author Response
Please see the attachment.

Round 2
Reviewer 3 Report
If editor and other reviewers agree I could also recommend the paper for publication after the revision.